# A Dirichlet Mixture Model of Hawkes Processes for Event Sequence Clustering

**Hongteng Xu**[*]
School of ECE
Georgia Institute of Technology
hongtengxu313@gmail.com

**Hongyuan Zha**
College of Computing
Georgia Institute of Technology
zha@cc.gatech.edu

## Abstract

How to cluster event sequences generated via different point processes is an interesting and important problem in statistical machine learning. To solve this problem, we propose and discuss an effective model-based clustering method based on a novel Dirichlet mixture model of a special but significant type of point processes — Hawkes process. The proposed model generates the event sequences with different clusters from the Hawkes processes with different parameters, and uses a Dirichlet distribution as the prior distribution of the clusters. We prove the identifiability of our mixture model and propose an effective variational Bayesian inference algorithm to learn our model. An adaptive inner iteration allocation strategy is designed to accelerate the convergence of our algorithm. Moreover, we investigate the sample complexity and the computational complexity of our learning algorithm in depth. Experiments on both synthetic and real-world data show that the clustering method based on our model can learn structural triggering patterns hidden in asynchronous event sequences robustly and achieve superior performance on clustering purity and consistency compared to existing methods.

## 1 Introduction

In many practical situations, we need to deal with a huge amount of irregular and asynchronous sequential data. Typical examples include the viewing records of users in an IPTV system, the electronic health records of patients in hospitals, among many others. All of these data are so-called *event sequences*, each of which contains a series of events with different types in the continuous time domain, e.g., when and which TV program a user watched, when and which care unit a patient is transferred to. Given a set of event sequences, an important task is learning their clustering structure robustly. Event sequence clustering is meaningful for many practical applications. Take the previous two examples: clustering IPTV users according to their viewing records is beneficial to the program recommendation system and the ads serving system; clustering patients according to their health records helps hospitals to optimize their medication resources.

Event sequence clustering is very challenging. Existing work mainly focuses on clustering synchronous (or aggregated) time series with discrete time-lagged observations [19, 23, 39]. Event sequences, on the contrary, are in the continuous time domain, so it is difficult to find a universal and tractable representation for them. A potential solution is constructing features of event sequences via parametric [22] or nonparametric [18] methods. However, these feature-based methods have a high risk of overfitting because of the large number of parameters. What is worse, these methods actually decompose the clustering problem into two phases: extracting features and learning clusters. As a result, their clustering results are very sensitive to the quality of learned (or predefined) features.

---

[*]Corresponding author.

To make concrete progress, we propose a **D**irichlet **M**ixture model of **H**awkes **P**rocesses (DMHP for short) and study its performance on event sequence clustering in depth. In this model, the event sequences belonging to different clusters are modeled via different Hawkes processes. The priors of the Hawkes processes' parameters are designed based on their physically-meaningful constraints. The prior of the clusters is generated via a Dirichlet distribution. We propose a variational Bayesian inference algorithm to learn the DMHP model in a nested Expectation-Maximization (EM) framework. In particular, we introduce a novel inner iteration allocation strategy into the algorithm with the help of open-loop control theory, which improves the convergence of the algorithm. We prove the local identifiability of our model and show that our learning algorithm has better sample complexity and computational complexity than its competitors.

The contributions of our work include: 1) We propose a novel Dirichlet mixture model of Hawkes processes and demonstrate its local identifiability. To our knowledge, it is the first systematical research on the identifiability problem in the task of event sequence clustering. 2) We apply an adaptive inner iteration allocation strategy based on open-loop control theory to our learning algorithm and show its superiority to other strategies. The proposed strategy achieves a trade-off between convergence performance and computational complexity. 3) We propose a DMHP-based clustering method. It requires few parameters and is robust to the problems of overfitting and model misspecification, which achieves encouraging clustering results.

## 2   Related Work

A temporal point process [4] is a random process whose realization consists of an event sequence $\{(t_i, c_i)\}_{i=1}^M$ with time stamps $t_i \in [0, T]$ and event types $c_i \in \mathcal{C} = \{1, ..., C\}$. It can be equivalently represented as $C$ counting processes $\{N_c(t)\}_{c=1}^C$, where $N_c(t)$ is the number of type-$c$ events occurring at or before time $t$. A way to characterize point processes is via the intensity function $\lambda_c(t) = \mathbb{E}[dN_c(t)|\mathcal{H}_t^{\mathcal{C}}]/dt$, where $\mathcal{H}_t^{\mathcal{C}} = \{(t_i, c_i)|t_i < t, c_i \in \mathcal{C}\}$ collects historical events of all types before time $t$. It is the expected instantaneous rate of happening type-$c$ events given the history, which captures the phenomena of interests, i.e., self-triggering [13] or self-correcting [44].

**Hawkes Processes.** A Hawkes process [13] is a kind of point processes modeling complicated event sequences in which historical events have influences on current and future ones. It can also be viewed as a cascade of non-homogeneous Poisson processes [8, 34]. We focus on the clustering problem of the event sequences obeying Hawkes processes because Hawkes processes have been proven to be useful for describing real-world data in many applications, e.g., financial analysis [1], social network analysis [3, 51], system analysis [22], and e-health [30, 42]. Hawkes processes have a particular form of intensity:

$$\lambda_c(t) = \mu_c + \sum_{c'=1}^C \int_0^t \phi_{cc'}(s)dN_{c'}(t-s), \tag{1}$$

where $\mu_c$ is the exogenous base intensity independent of the history while $\sum_{c'=1}^C \int_0^t \phi_{cc'}(s)dN_{c'}(t-s)$ the endogenous intensity capturing the peer influence. The decay in the influence of historical type-$c'$ events on the subsequent type-$c$ events is captured via the so-called *impact function* $\phi_{cc'}(t)$, which is nonnegative. A lot of existing work uses predefined impact functions with known parameters, e.g., the exponential functions in [29, 50] and the power-law functions in [49]. To enhance the flexibility, a nonparametric model of 1-D Hawkes process was first proposed in [16] based on ordinary differential equation (ODE) and extended to multi-dimensional case in [22, 51]. Another nonparametric model is the contrast function-based model in [30], which leads to a Least-Squares (LS) problem [7]. A Bayesian nonparametric model combining Hawkes processes with infinite relational model is proposed in [3]. Recently, the basis representation of impact functions was used in [6, 15, 41] to avoid discretization.

**Sequential Data Clustering and Mixture Models.** Traditional methods mainly focus on clustering synchronous (or aggregated) time series with discrete time-lagged variables [19, 23, 39]. These methods rely on probabilistic mixture models [46], extracting features from sequential data and then learning clusters via a Gaussian mixture model (GMM) [25, 28]. Recently, a mixture model of Markov chains is proposed in [21], which learns potential clusters from aggregate data. For asynchronous event sequences, most of the existing clustering methods can be categorized into feature-based methods, clustering event sequences from learned or predefined features. Typical examples

include the Gaussian process-base multi-task learning method in [18] and the multi-task multi-dimensional Hawkes processes in [22]. Focusing on Hawkes processes, the feature-based mixture models in [5, 17, 47] combine Hawkes processes with Dirichlet processes [2, 36]. However, these methods aim at modeling clusters of events or topics hidden in event sequences (i.e., sub-sequence clustering), which cannot learn clusters of event sequences. To our knowledge, the model-based clustering method for event sequences has been rarely considered.

## 3  Proposed Model

### 3.1  Dirichlet Mixture Model of Hawkes Processes

Given a set of event sequences $\boldsymbol{S} = \{\boldsymbol{s}_n\}_{n=1}^N$, where $\boldsymbol{s}_n = \{(t_i, c_i)\}_{i=1}^{M_n}$ contains a series of events $c_i \in \mathcal{C} = \{1, ..., C\}$ and their time stamps $t_i \in [0, T_n]$, we model them via a mixture model of Hawkes processes. According to the definition of Hawkes process in (1), for the event sequence belonging to the $k$-th cluster its intensity function of type-$c$ event at time $t$ is

$$\lambda_c^k(t) = \mu_c^k + \sum_{t_i < t} \phi_{cc_i}^k(t - t_i) = \mu_c^k + \sum_{t_i < t} \sum_{d=1}^D a_{cc_i d}^k g_d(t - t_i), \qquad (2)$$

where $\boldsymbol{\mu}^k = [\mu_c^k] \in \mathbb{R}_+^C$ is the exogenous base intensity of the $k$-th Hawkes process. Following the work in [41], we represent each impact function $\phi_{cc'}^k(t)$ via basis functions as $\sum_d a_{cc'd}^k g_d(t - t_i)$, where $g_d(t) \geq 0$ is the $d$-th basis function and $\boldsymbol{A}^k = [a_{cc'd}^k] \in \mathbb{R}_{0+}^{C \times C \times D}$ is the coefficient tensor. Here we use Gaussian basis function, and their number $D$ can be decided automatically using the basis selection method in [41].

In our mixture model, the probability of the appearance of an event sequence $\boldsymbol{s}$ is

$$p(\boldsymbol{s}; \boldsymbol{\Theta}) = \sum_k \pi^k \text{HP}(\boldsymbol{s}|\boldsymbol{\mu}^k, \boldsymbol{A}^k), \ \text{HP}(\boldsymbol{s}|\boldsymbol{\mu}^k, \boldsymbol{A}^k) = \prod_i \lambda_{c_i}^k(t_i) \exp\Big(-\sum_c \int_0^T \lambda_c^k(s)ds\Big). \quad (3)$$

Here $\pi^k$'s are the probabilities of clusters and $\text{HP}(\boldsymbol{s}|\boldsymbol{\mu}^k, \boldsymbol{A}^k)$ is the conditional probability of the event sequence $\boldsymbol{s}$ given the $k$-th Hawkes process, which follows the intensity function-based definition in [4]. According to the Bayesian graphical model, we regard the parameters of Hawkes processes, $\{\boldsymbol{\mu}^k, \boldsymbol{A}^k\}$, as random variables. For $\boldsymbol{\mu}^k$'s, we consider its positiveness and assume that they obey $C \times K$ independent Rayleigh distributions. For $\boldsymbol{A}^k$'s, we consider its nonnegativeness and sparsity as the work in [22, 41, 50]) did, and assume that they obey $C \times C \times D \times K$ independent exponential distributions. The prior of cluster is a Dirichlet distribution. Therefore, we can describe the proposed Dirichlet mixture model of Hawkes process in a generative way as

$$\boldsymbol{\pi} \sim \text{Dir}(\alpha/K, ..., \alpha/K), \ k|\boldsymbol{\pi} \sim \text{Category}(\boldsymbol{\pi}),$$
$$\boldsymbol{\mu} \sim \text{Rayleigh}(\boldsymbol{B}), \ \boldsymbol{A} \sim \text{Exp}(\boldsymbol{\Sigma}), \ \boldsymbol{s}|k, \boldsymbol{\mu}, \boldsymbol{A} \sim \text{HP}(\boldsymbol{\mu}_k, \boldsymbol{A}_k),$$

Here $\boldsymbol{\mu} = [\mu_c^k] \in \mathbb{R}_+^{C \times K}$ and $\boldsymbol{A} = [a_{cc'd}^k] \in \mathbb{R}_{0+}^{C \times C \times D \times K}$ are parameters of Hawkes processes, and $\{\boldsymbol{B} = [\beta_c^k], \boldsymbol{\Sigma} = [\sigma_{cc'd}^k]\}$ are hyper-parameters. Denote the latent variables indicating the labels of clusters as matrix $\boldsymbol{Z} \in \{0, 1\}^{N \times K}$. We can factorize the joint distribution of all variables as[2]

$$p(\boldsymbol{S}, \boldsymbol{Z}, \boldsymbol{\pi}, \boldsymbol{\mu}, \boldsymbol{A}) = p(\boldsymbol{S}|\boldsymbol{Z}, \boldsymbol{\mu}, \boldsymbol{A})p(\boldsymbol{Z}|\boldsymbol{\pi})p(\boldsymbol{\pi})p(\boldsymbol{\mu})p(\boldsymbol{A}), \ \text{where}$$
$$p(\boldsymbol{S}|\boldsymbol{Z}, \boldsymbol{\mu}, \boldsymbol{A}) = \prod_{n,k} \text{HP}(\boldsymbol{s}_n|\boldsymbol{\mu}^k, \boldsymbol{A}^k)^{z_{nk}}, \quad p(\boldsymbol{Z}|\boldsymbol{\pi}) = \prod_{n,k} (\pi^k)^{z_{nk}}, \qquad (4)$$
$$p(\boldsymbol{\pi}) = \text{Dir}(\boldsymbol{\pi}|\boldsymbol{\alpha}), \quad p(\boldsymbol{\mu}) = \prod_{c,k} \text{Rayleigh}(\mu_c^k|\beta_c^k), \quad p(\boldsymbol{A}) = \prod_{c,c',d,k} \text{Exp}(a_{cc'd}^k|\sigma_{cc'd}^k).$$

Our mixture model of Hawkes processes are different from the models in [5, 17, 47]. Those models focus on the sub-sequence clustering problem within an event sequence. The intensity function is a weighted sum of multiple intensity functions of different Hawkes processes. Our model, however, aims at finding the clustering structure across different sequences. The intensity of each event is generated via a single Hawkes process, while the likelihood of an event sequence is a mixture of likelihood functions from different Hawkes processes.

## 3.2 Local Identifiability

One of the most important questions about our mixture model is whether it is identifiable or not. According to the definition of Hawkes process and the work in [26, 31], we can prove that our model is locally identifiable. The proof of the following theorem is given in the supplementary file.

**Theorem 3.1.** *When the time of observation goes to infinity, the mixture model of the Hawkes processes defined in (3) is locally identifiable, i.e., for each parameter point* $\Theta = vec\left(\begin{bmatrix} \pi^1 & \cdots & \pi^K \\ \theta^1 & \cdots & \theta^K \end{bmatrix}\right)$, *where* $\theta^k = \{\mu^k, A^k\} \in \mathbb{R}_+^C \times \mathbb{R}_{0+}^{C \times C \times D}$ *for* $k = 1, .., K$, *there exists an open neighborhood of* $\Theta$ *containing no other* $\Theta'$ *which makes* $p(s; \Theta) = p(s; \Theta')$ *holds for all possible* $s$.

# 4 Proposed Learning Algorithm

## 4.1 Variational Bayesian Inference

Instead of using purely MCMC-based learning method like [29], we propose an effective variational Bayesian inference algorithm to learn (4) in a nested EM framework. Specifically, we consider a variational distribution having the following factorization:

$$q(Z, \pi, \mu, A) = q(Z)q(\pi, \mu, A) = q(Z)q(\pi)\prod_k q(\mu^k)q(A^k). \tag{5}$$

An EM algorithm can be used to optimize (5).

**Update Responsibility (E-step).** The logarithm of the optimized factor $q^*(Z)$ is approximated as

$$\log q^*(Z) = \mathbb{E}_\pi[\log p(Z|\pi)] + \mathbb{E}_{\mu,A}[\log p(S|Z, \mu, A)] + \mathsf{C}$$

$$= \sum_{n,k} z_{nk}\left(\mathbb{E}[\log \pi^k] + \mathbb{E}[\log \mathrm{HP}(s_n|\mu^k, A^k)]\right) + \mathsf{C}$$

$$= \sum_{n,k} z_{nk}\left(\mathbb{E}[\log \pi^k] + \mathbb{E}[\sum_i \log \lambda_{c_i}^k(t_i) - \sum_c \int_0^{T_n} \lambda_c^k(s)ds]\right) + \mathsf{C}$$

$$\approx \sum_{n,k} z_{nk}\underbrace{\left(\mathbb{E}[\log \pi^k] + \sum_i \left(\log \mathbb{E}[\lambda_{c_i}^k(t_i)] - \frac{\mathrm{Var}[\lambda_{c_i}^k(t_i)]}{2\mathbb{E}^2[\lambda_{c_i}^k(t_i)]}\right) - \sum_c \mathbb{E}[\int_0^{T_n} \lambda_c^k(s)ds]\right)}_{\rho_{nk}} + \mathsf{C}.$$

where $\mathsf{C}$ is a constant and $\mathrm{Var}[\cdot]$ represents the variance of random variable. Each term $\mathbb{E}[\log \lambda_c^k(t)]$ is approximated via its second-order Taylor expansion $\log \mathbb{E}[\lambda_c^k(t)] - \frac{\mathrm{Var}[\lambda_c^k(t)]}{2\mathbb{E}^2[\lambda_c^k(t)]}$ [37]. Then, the responsibility $r_{nk}$ is calculated as

$$r_{nk} = \mathbb{E}[z_{nk}] = \rho_{nk}/(\sum_j \rho_{nj}). \tag{6}$$

Denote $N_k = \sum_n r_{nk}$ for all $k$'s.

**Update Parameters (M-step).** The logarithm of optimal factor $q^*(\pi, \mu, A)$ is

$$\log q^*(\pi, \mu, A)$$

$$= \sum_k \log(p(\mu^k)p(A^k)) + \mathbb{E}_Z[\log p(Z|\pi)] + \log p(\pi) + \sum_{n,k} r_{nk}\log \mathrm{HP}(s_n|\mu^k, A^k) + \mathsf{C}.$$

We can estimate the parameters of Hawkes processes via:

$$\hat{\mu}, \widehat{A} = \arg \max_{\mu, A} \log(p(\mu)p(A)) + \sum_{n,k} r_{nk}\log \mathrm{HP}(s_n|\mu^k, A^k). \tag{7}$$

Following the work in [41, 47, 50], we need to apply an EM algorithm to solve (7) iteratively. After getting optimal $\hat{\mu}$ and $\widehat{A}$, we update distributions as

$$\Sigma^k = \widehat{A}^k, \ B^k = \sqrt{2/\pi}\hat{\mu}^k. \tag{8}$$

**Update The Number of Clusters** $K$. When the number of clusters $K$ is unknown, we initialize $K$ randomly and update it in the learning phase. There are multiple methods to update the number of

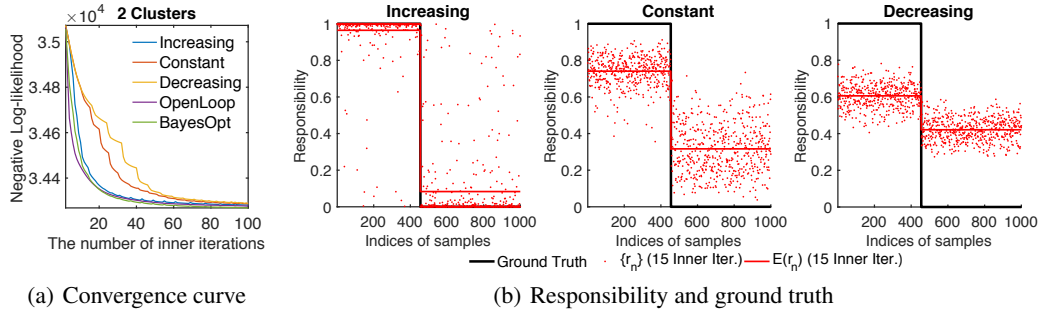

(a) Convergence curve          (b) Responsibility and ground truth

Figure 1: The data contain 200 event sequences generated via two 5-dimensional Hawkes processes. (a) Each curve is the average of 5 trials' results. In each trial, total 100 inner iterations are applied. The increasing (decreasing) strategy changes the number of inner iterations from 2 to 8 (from 8 to 2). The constant strategy fixes the number to 5. (b) The black line is the ground truth. The red dots are responsibilities after 15 inner iterations, and the red line is their average.

clusters. Regrading our Dirichlet distribution as a finite approximation of a Dirichlet process, we set a large initial $K$ as the truncation level. A simple empirical method is discarding the empty cluster (i.e., $N_k = 0$) and merging the cluster with $N_k$ smaller than a threshold $N_{min}$ in the learning phase. Besides this, we can apply the MCMC in [11, 48] to update $K$ via merging or splitting clusters.

Repeating the three steps above, our algorithm maximizes the log-likelihood function (i.e., the logarithm of (4)) and achieves optimal $\{\boldsymbol{\Sigma}, \boldsymbol{B}\}$ accordingly. Both the details of our algorithm and its computational complexity are given in the supplementary file.

## 4.2 Inner Iteration Allocation Strategy and Convergence Analysis

Our algorithm is in a nested EM framework, where the outer iteration corresponds to the loop of E-step and M-step and the inner iteration corresponds to the inner EM in the M-step. The runtime of our algorithm is linearly proportional to the total number of inner iterations. Given fixed runtime (or the total number of inner iterations), both the final achievable log-likelihood and convergence behavior of the algorithm highly depend on how we allocate the number of inner iterations across the outer iterations. In this work, we test three inner iteration allocation strategies. The first strategy is *heuristic*, which fixes, increases, or decreases the number of inner iterations as the outer iteration progresses. Compared with constant inner iteration strategy, the increasing or decreasing strategy might improve the convergence of algorithm [9]. The second strategy is based on *open-loop control* [27]: in each outer iteration, we compute objective function via two methods respectively — updating parameters directly (i.e., continuing current M-step and going to next inner iteration) or first updating responsibilities and then updating parameters (i.e., going to a new loop of E-step and M-step and starting a new outer iteration). The parameters corresponding to the smaller negative log-likelihood are preserved. The third strategy is applying *Bayesian optimization* [33, 35] to optimize the number of inner iterations per outer iteration via maximizing the expected improvement.

We apply these strategies to a synthetic data set and visualize their impacts on the convergence of our algorithm in Fig. 1(a). The open-loop control strategy and the Bayesian optimization strategy obtain comparable performance on the convergence of algorithm. They outperform heuristic strategies (i.e., increasing, decreasing and fixing the number of inner iterations per outer iteration), which reduce the negative log-likelihood more rapidly and reach lower value finally. Although adjusting the number of inner iterations via different methodologies, both these two strategies tend to increase the number of inner iterations w.r.t. the number of outer iterations. In the beginning of algorithm, the open-loop control strategy updates responsibilities frequently, and similarly, the Bayesian optimization strategy assigns small number of inner iterations. The heuristic strategy that increasing the number of inner iterations follows the same tendency, and therefore, is just slightly worse than the open-loop control and the Bayesian optimization. This phenomenon is because the estimated responsibility is not reliable in the beginning. Too many inner iterations at that time might make learning results fall into bad local optimums.

Fig. 1(b) further verifies our explanation. With the help of the increasing strategy, most of the responsibilities converge to the ground truth with high confidence after just 15 inner iterations, because

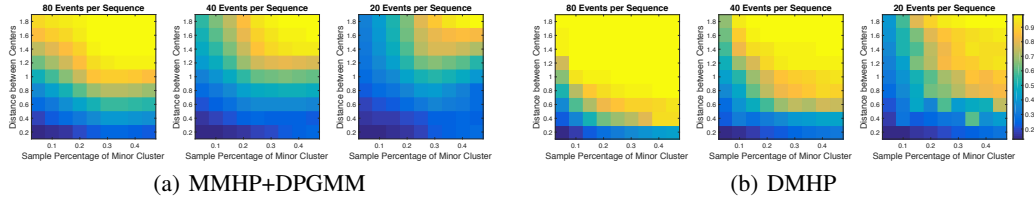

Figure 2: Comparisons for various methods on F1 score of minor cluster.

the responsibilities has been updated over 5 times. On the contrary, the responsibilities corresponding to the constant and the decreasing strategies have more uncertainty — many responsibilities are around 0.5 and far from the ground truth.

Based on the analysis above, the *increasing* allocation strategy indeed improves the convergence of our algorithm, and the open-loop control and the Bayesian optimization are superior to other competitors. Because the computational complexity of the open-loop control is much lower than that of the Bayesian optimization, in the following experiments, we apply open-loop control strategy to our learning algorithm. The scheme of our learning algorithm and more detailed convergence analysis can be found in the supplementary file.

### 4.3 Empirical Analysis of Sample Complexity

Focusing on the task of clustering event sequences, we investigate the sample complexity of our DMHP model and its learning algorithm. In particular, we want to show that the clustering method based on our model requires fewer samples than existing methods to identify clusters successfully. Among existing methods, the main competitor of our method is the clustering method based on the multi-task multi-dimensional Hawkes process (MMHP) model in [22]. It learns a specific Hawkes process for each sequence and clusters the sequences via applying the Dirichlet processes Gaussian mixture model (DPGMM) [10, 28] to the parameters of the corresponding Hawkes processes.

Following the work in [14], we demonstrate the superiority of our DMHP-based clustering method through the comparison on the identifiability of minor clusters given finite number of samples. Specifically, we consider a binary clustering problem with $500$ event sequences. For the $k$-th cluster, $k = 1, 2$, $N_k$ event sequences are generated via a 1-dimensional Hawkes processes with parameter $\boldsymbol{\theta}^k = \{\boldsymbol{\mu}^k, \boldsymbol{A}^k\}$. Taking the parameter as a representation of the clustering center, we can calculate the distance between two clusters as $d = \|\boldsymbol{\theta}^1 - \boldsymbol{\theta}^2\|_2$. Assume that $N_1 < N_2$, we denote the first cluster as "minor" cluster, whose sample percentage is $\pi^1 = \frac{N_1}{N_1 + N_2}$. Applying our DMHP model and its learning algorithm to the data generated with different $d$'s and $\pi^1$'s, we can calculate the F1 scores of the minor cluster w.r.t. $\{d, \pi\}$. The high F1 score means that the minor cluster is identified with high accuracy. Fig. 2 visualizes the maps of F1 scores generated via different methods w.r.t. the number of events per sequence. We can find that the F1 score obtained via our DMHP-based method is close to $1$ in most situations. Its identifiable area (yellow part) is much larger than that of the MMHP+DPGMM method consistently w.r.t. the number of events per sequence. The unidentifiable cases happen only in the following two situations: the parameters of different clusters are nearly equal (i.e., $d \to 0$); or the minor cluster is extremely small (i.e., $\pi^1 \to 0$). The enlarged version of Fig. 2 is given in the supplementary file.

## 5 Experiments

To demonstrate the feasibility and the efficiency of our **DMHP**-based sequence clustering method, we compare it with the state-of-the-art methods, including the vector auto-regressive (**VAR**) method [12], the Least-Squares (**LS**) method in [7], and the multi-task multi-dimensional Hawkes process (**MMHP**) in [22]. All of the three competitors first learn features of sequences and then apply the **DPGMM** [10] to cluster sequences. The VAR discretizes asynchronous event sequences to time series and learns transition matrices as features. Both the LS and the MMHP learn a specific Hawkes process for each event sequence. For each event sequence, we calculate its infectivity matrix $\boldsymbol{\Phi} = [\phi_{cc'}]$, where the element $\phi_{cc'}$ is the integration of impact function (i.e., $\int_0^\infty \phi_{cc'}(t)dt$), and use it as the feature.

Table 1: Clustering Purity on Synthetic Data.

| $C$ | $K$ | Sine-like $\phi(t)$ | | | | Piecewise constant $\phi(t)$ | | | |
|---|---|---|---|---|---|---|---|---|---|
| | | VAR+ DPGMM | LS+ DPGMM | MMHP+ DPGMM | DMHP | VAR+ DPGMM | LS+ DPGMM | MMHP+ DPGMM | DMHP |
| 5 | 2 | 0.5235 | 0.5639 | 0.5917 | **0.9898** | 0.5222 | 0.5589 | 0.5913 | **0.8085** |
| | 3 | 0.3860 | 0.5278 | 0.5565 | **0.9683** | 0.3618 | 0.4402 | 0.4517 | **0.7715** |
| | 4 | 0.2894 | 0.4365 | 0.5112 | **0.9360** | 0.2901 | 0.3365 | 0.3876 | **0.7056** |
| | 5 | 0.2543 | 0.3980 | 0.4656 | **0.9055** | 0.2476 | 0.2980 | 0.3245 | **0.6774** |

For the synthetic data with clustering labels, we use *clustering purity* [24] to evaluate various methods:

$$\text{Purity} = \frac{1}{N} \sum_{k=1}^{K} \max_{j \in \{1,\ldots,K'\}} |\mathcal{W}_k \cap \mathcal{C}_j|,$$

where $\mathcal{W}_k$ is the learned index set of sequences belonging to the $k$-th cluster, $\mathcal{C}_j$ is the real index set of sequence belonging to the $j$-th class, and $N$ is the total number of sequences. For the real-world data, we visualize the infectivity matrix of each cluster and measure the *clustering consistency* via a cross-validation method [38, 40]. The principle is simple: because random sampling does not change the clustering structure of data, a clustering method with high consistency should preserve the pairwise relationships of samples in different trials. Specifically, we test each clustering method with $J \ (= 100)$ trials. In the $j$-th trial, data is randomly divided into two folds. After learning the corresponding model from the training fold, we apply the method to the testing fold. We enumerate all pairs of sequences within a same cluster in the $j$-th trial and count the pairs preserved in all other trials. The clustering consistency is the minimum proportion of preserved pairs over all trials:

$$\text{Consistency} = \min_{j \in \{1,\ldots,J\}} \sum_{j' \neq j} \sum_{(n,n') \in \mathcal{M}_j} \frac{1\{k_n^{j'} = k_{n'}^{j'}\}}{(J-1)|\mathcal{M}_j|},$$

where $\mathcal{M}_j = \{(n, n')|k_n^j = k_{n'}^j\}$ is the set of sequence pairs within same cluster in the $j$-th trial, and $k_n^j$ is the index of cluster of the $n$-th sequence in the $j$-th trial.

## 5.1 Synthetic Data

We generate two synthetic data sets with various clusters using sine-like impact functions and piecewise constant impact functions respectively. In each data set, the number of clusters is set from 2 to 5. Each cluster contains 400 event sequences, and each event sequence contains 50 $(= M_n)$ events and 5 $(= C)$ event types. The elements of exogenous base intensity are sampled uniformly from $[0, 1]$. Each sine-like impact function in the $k$-th cluster is formulated as $\phi_{cc'}^k = b_{cc'}^k(1 - \cos(\omega_{cc'}^k(t - s_{cc'}^k)))$, where $\{b_{cc'}^k, \omega_{cc'}^k, s_{cc'}^k\}$ are sampled randomly from $[\frac{\pi}{5}, \frac{2\pi}{5}]$. Each piecewise constant impact function is the truncation of the corresponding sine-like impact function, i.e., $2b_{cc'}^k \times \text{round}(\phi_{cc'}^k/(2b_{cc'}^k))$.

Table 1 shows the clustering purity of various methods on the synthetic data. Compared with the three competitors, our DMHP obtains much better clustering purity consistently. The VAR simply treats asynchronous event sequences as time series, which loses the information like the order of events and the time delay of adjacent events. Both the LS and the MMHP learn Hawkes process for each individual sequence, which might suffer to over-fitting problem in the case having few events per sequence. These competitors decompose sequence clustering into two phases: learning feature and applying DPGMM, which is very sensitive to the quality of feature. The potential problems above lead to unsatisfying clustering results. Our DMHP method, however, is model-based, which learns clustering result directly and reduces the number of unknown variables greatly. As a result, our method avoids the problems of these three competitors and obtains superior clustering results. Additionally, the learning results of the synthetic data with piecewise constant impact functions prove that our DMHP method is relatively robust to the problem of model misspecification — although our Gaussian basis cannot fit piecewise constant impact functions well, our method still outperforms other methods greatly.

## 5.2 Real-world Data

We test our clustering method on two real-world data sets. The first is the ICU patient flow data used in [43], which is extracted from the MIMIC II data set [32]. This data set contains the transition

Table 2: Clustering Consistency on Real-world Data.

| Method | VAR+DPGMM | LS+DPGMM | MMHP+DPGMM | DMHP |
|---|---|---|---|---|
| ICU Patient | 0.0901 | 0.1390 | 0.3313 | **0.3778** |
| IPTV User | 0.0443 | 0.0389 | 0.1382 | **0.2004** |

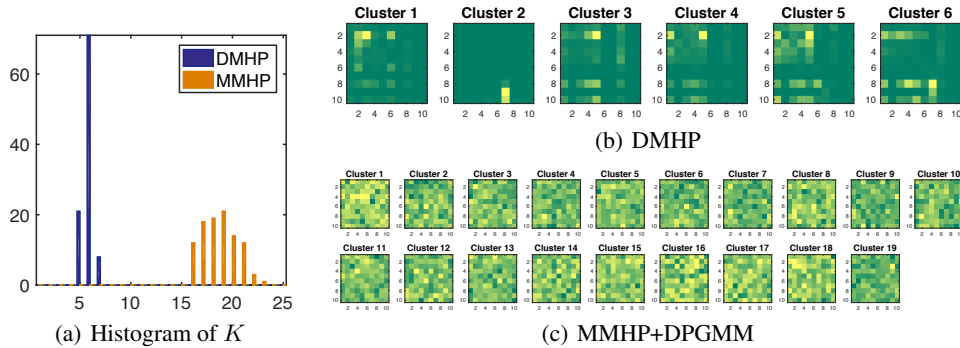

(a) Histogram of $K$

(b) DMHP

(c) MMHP+DPGMM

Figure 3: Comparisons on the ICU patient flow data.

processes of $30,308$ patients among different kinds of care units. The patients can be clustered according to their transition processes. The second is the IPTV data set in [20, 22], which contains $7,100$ IPTV users' viewing records collected via Shanghai Telecomm Inc. The TV programs are categorized into 16 classes and the viewing behaviors more than 20 minutes are recorded. Similarly, the users can be clustered according to their viewing records. The event sequences in these two data have strong but structural triggering patterns, which can be modeled via different Hawkes processes.

Table 2 shows the performance of various clustering methods on the clustering consistency. We can find that our method outperforms other methods obviously, which means that the clustering result obtained via our method is more stable and consistent than other methods' results. In Fig. 3 we visualize the comparison for our method and its main competitor MMHP+DPGMM on the ICU patient flow data. Fig. 3(a) shows the histograms of the number of clusters for the two methods. We can find that MMHP+DPGMM method tends to over-segment data into too many clusters. Our DMHP method, however, can find more compact clustering structure. The distribution of the number of clusters concentrates to 6 and 19 for the two data sets, respectively. In our opinion, this phenomenon reflects the drawback of the feature-based method — the clustering performance is highly dependent on the quality of feature while the clustering structure is not considered sufficiently in the phase of extracting feature. Taking learned infectivity matrices as representations of clusters, we compare our DMHP method with MMHP+DPGMM in Figs. 3(b) and 3(c). The infectivity matrices obtained by our DMHP are sparse and with distinguishable structure, while those obtained by MMHP+DPGMM are chaotic — although MMHP also applies sparse regularizer to each event sequence' infectivity matrix, it cannot guarantee the average of the infectivity matrices in a cluster is still sparse. Same phenomena can also be observed in the experiments on the IPTV data. More experimental results are given in the supplementary file.

## 6 Conclusion and Future Work

In this paper, we propose and discuss a Dirichlet mixture model of Hawkes processes and achieve a model-based solution to event sequence clustering. We prove the identifiability of our model and analyze the convergence, sample complexity and computational complexity of our learning algorithm. In the aspect of methodology, we plan to study other potential priors, e.g., the prior based on determinantial point processes (DPP) in [45], to improve the estimation of the number of clusters, and further accelerate our learning algorithm via optimizing inner iteration allocation strategy in near future. Additionally, our model can be extended to Dirichlet process mixture model when $K \to \infty$. In that case, we plan to apply Bayesian nonparametrics to develop new learning algorithms. The source code can be found at `https://github.com/HongtengXu/Hawkes-Process-Toolkit`.

## Acknowledgment

This work is supported in part by NSF IIS-1639792, IIS-1717916, and CMMI-1745382.

## Footnotes

[2]Rayleigh$(x|\beta) = \frac{x}{\beta^2} e^{-\frac{x^2}{2\beta^2}}$, Exp$(x|\sigma) = \frac{1}{\sigma} e^{-\frac{x}{\sigma}}$, $x \geq 0$.

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
