[Supplementary Material · nips_2017_supp.pdf]

# A Dirichlet Mixture Model of Hawkes Processes for Event Sequence Clustering — Supplementary File

**Hongteng Xu**[*]
School of ECE
Georgia Institute of Technology
hongtengxu313@gmail.com

**Hongyuan Zha**
College of Computing
Georgia Institute of Technology
zha@cc.gatech.edu

## 1 The Proof of Local Identifiability

Before proving the local identifiability of our DMHP model, we first introduce some key concepts. A temporal point process is a random process whose realization consists of a list of discrete events in time $\{t_i\}$ with $t_i \in [0, T]$. Here $[0, T]$ is the time interval of the process. It can be equivalently represented as a counting process, $N = \{N(t)|t \in [0, T]\}$, where $N(t)$ records the number of events before time $t$. A multi-dimensional point process with $C$ types of event is represented by $C$ counting processes $\{N_c\}_{c=1}^C$ on a probability space $(\Omega, \mathfrak{F}, \mathbb{P})$. $N_c = \{N_c(t)|t \in [0, T]\}$, where $N_c(t)$ is the number of type-$c$ events occurring at or before time $t$. $\Omega = [0, T] \times \mathcal{C}$ is the sample space. $\mathcal{C} = \{1, ..., C\}$ is the set of event types. $\mathfrak{F} = (\mathfrak{F}(t))_{t \in \mathbb{R}}$ is the filtration representing the set of events sequence the process can realize until time $t$. $\mathbb{P}$ is the probability measure.

Hawkes process is a kind of temporal point processes having self-and mutually-triggering patterns. The triggering of historical events on current ones in a Hawkes process can be modeled as branch processes [1, 7]. As a result, Hawkes Process can be represented as a superposition of many non-homogeneous Poisson process. Due to the superposition theorem of Poisson processes, the superposition of the individual processes is equivalent to the point process with summation of their intensity function. Given this we can break the counting process associated to each addition to the intensity function (or associated to each event): $N(t) = \sum_{i=0}^n N^i(t)$, where $N^0(t)$ is the counting process associated to the baseline intensity $\mu(t)$ and $N^i(t)$ is the non-homgenous Poisson process for the $i$-th branch. Similarly, we can write the intensity function of Hawkes process as $\lambda(t) = \sum_{i=0}^n \lambda^i(t)$, where $\lambda^i(t)$ is the intensity of the $i$-th branch.

**Definition 1.1.** *Two parameter points $\Theta^1$ and $\Theta^2$ are said to be observationally equivalent if $p(\boldsymbol{s}; \Theta^1) = p(\boldsymbol{s}; \Theta^2)$ for all samples $\boldsymbol{s}$'s in sample space.*

**Definition 1.2.** *A parameter point $\Theta^0$ is said to be locally identifiable if there exists an open neighborhood of $\Theta^0$ containing no other $\Theta$ in the parameter space which is observationally equivalent.*

**Definition 1.3.** *Let $\boldsymbol{I}(\Theta)$ be a matrix whose elements are continuous functions of $\Theta$ everywhere in the parameter space. The point $\Theta^0$ is said to be a regular point of the matrix if there exists an open neighborhood of $\Theta^0$ in which $\boldsymbol{I}(\Theta)$ has constant rank.*

The information matrix $\boldsymbol{I}(\Theta)$ is defined as

$$\boldsymbol{I}(\Theta) = \mathbb{E}_{\boldsymbol{s}} \left[ \frac{\partial \log p(\boldsymbol{s}; \Theta)}{\partial \Theta} \frac{\partial \log p(\boldsymbol{s}; \Theta)}{\partial \Theta^\top} \right] = \mathbb{E}_{\boldsymbol{s}} \left[ \frac{1}{p^2(\boldsymbol{s}; \Theta)} \frac{\partial p(\boldsymbol{s}; \Theta)}{\partial \Theta} \frac{\partial p(\boldsymbol{s}; \Theta)}{\partial \Theta^\top} \right],$$

The local identifiability of our DMHP model is based on the following two theorems.

**Theorem 1.1.** *[4] The information matrix $\boldsymbol{I}(\Theta)$ is positive definite if and only if there does not exist a nonzero vector of constants $\boldsymbol{w}$ such that $\boldsymbol{w}^\top \frac{\partial p(\boldsymbol{s}; \Theta)}{\partial \Theta} = 0$ for all samples $\boldsymbol{s}$'s in sample space.*

---

[*]Corresponding author.

**Theorem 1.2.** *[5] Let $\Theta^0$ be a regular point of the information matrix $I(\Theta)$. Then $\Theta^0$ is locally identifiable if and only if $I(\Theta^0)$ is nonsingular.*

To our DMHP model, the log-likelihood function is composed with differentiable functions of $\Theta$. Therefore, the elements of information matrix $I(\Theta)$ are continuous functions w.r.t. $\Theta$ in the parameter space. According to Theorems 1.1 and 1.2, our Theorem holds if and only if to each vector $\frac{\partial p(s;\Theta)}{\partial \Theta}$ w.r.t. a point $\Theta$, there does not exist a nonzero vector of constants $w$ such that $w^\top \frac{\partial p(s;\Theta)}{\partial \Theta} = 0$ for all event sequences $s \in \mathfrak{F}$.

Assume that there exists a nonzero $w$ such that $w^\top \frac{\partial p(s;\Theta)}{\partial \Theta} = 0$ for all $s \in \mathfrak{F}$. We have the following **counter-evidence:** Considering the simplest case — the mixture of two Poisson processes (or equivalently, two 1-dimensional Hawkes processes whose impact functions $\phi(t) \equiv 0$), we can write its likelihood given a sequence with $N$ events in $[0, T]$ as

$$p(s_N; \Theta) = \pi \lambda_1^N \exp(-T\lambda_1) + (1 - \pi)\lambda_2^N \exp(-T\lambda_2) = \Lambda_1 + \Lambda_2,$$

where $\Theta = [\pi, \lambda_1, \lambda_2]^\top$, $\lambda_1 \neq \lambda_2$. According to our assumption, we have

$$w^\top \frac{\partial p(s_N; \Theta)}{\partial \Theta} = w^\top \begin{bmatrix} \frac{\Lambda_1}{\pi} - \frac{\Lambda_2}{1-\pi} \\ (\frac{N}{\lambda_1} - T)\Lambda_1 \\ (\frac{N}{\lambda_2} - T)\Lambda_2 \end{bmatrix} = 0,$$

Denote the time stamp of the last event as $t_N$, we can generate new event sequences $\{s_{N+n}\}_{n=1}^\infty$ via adding $n$ events in $(t_N, T]$, and

$$w^\top \frac{\partial p(s_{N+n}; \Theta)}{\partial \Theta} = w^\top \begin{bmatrix} \lambda_1^n \frac{\Lambda_1}{\pi} - \lambda_2^n \frac{\Lambda_2}{1-\pi} \\ ((N + n) - T\lambda_1)\lambda_1^{n-1}\Lambda_1 \\ ((N + n) - T\lambda_2)\lambda_2^{n-1}\Lambda_2 \end{bmatrix}.$$

$w^\top \frac{\partial p(s_{N+n};\Theta)}{\partial \Theta} = 0$ for $n = 0, ..., \infty$ requires $w \equiv 0$ or all $\frac{\partial p(s_{N+n};\Theta)}{\partial \Theta}$ are coplanar. However, according to the formulation above, for arbitrary three different $n_1, n_2, n_3 \in \{0, ..., \infty\}$, $\sum_{i=1}^3 \alpha_i \frac{\partial p(s_{N+n_i};\Theta)}{\partial \Theta} = 0$ holds if and only if $\alpha_1 = \alpha_2 = \alpha_3 = 0$.[2] Therefore, $w \equiv 0$, which violates the assumption above.

Such a counter-evidence can also be found in more general case, i.e., mixtures of multiple multi-dimensional Hawkes processes because Hawkes process is a superposition of many non-homogeneous Poisson process. As a result, according to Theorems 1.1 and 1.2, each point $\Theta$ in the parameter space is regular point of $I(\Theta)$ and the $I(\Theta)$ is nonsingular, and thus, our DMHP model is locally identifiable.

## 2 The Selection of Basis Functions

In our work, we apply Gaussian basis functions to our model. We use the basis selection method in [9] to decide the bandwidth and the number of basis functions. In particular, we focus on the impact functions having Fourier transformation. The representation of impact function, i.e., $\phi_{cc'}(t) = \sum_{d=1}^D a_{cc'}^d g_d(t)$, can be explained as a sampling process, where $\{a_{cc'}^d\}_{d=1}^D$ can be viewed as the discretized samples of $\phi_{cc'}(t)$ in $[0, T]$ and each $g_d(t) = \kappa_\omega(t, t_d)$ is sampling function with cut-off frequence $\omega$ and center $t_d$. Given training sequences $S = \{s_n = \{(t_i, c_i)\}_{i=1}^{M_n}\}_{n=1}^N$, we can estimate $\lambda(t)$ empirically via a Gaussian-based kernel density estimator:

$$\lambda(t) = \sum_{n=1}^N \sum_{i=1}^{M_n} G_h(t - t_i). \tag{1}$$

Here $G_h(t - t_i) = \exp(-\frac{(t-t_i)^2}{2h^2})$ is a Gaussian kernel with the bandwidth $h$. Instead of computing (1), we directly apply Silverman's rule of thumb [6] to set optimal $h = (\frac{4\hat{\sigma}^5}{3\sum_n M_n})^{0.2}$, where $\hat{\sigma}$ is the standard deviation of time stamps $\{t_i\}$. Applying Fourier transform, we compute an upper bound for

the spectral of $\lambda(t)$ as

$$
|\hat{\lambda}(\omega)| = \left| \int_{-\infty}^{\infty} \lambda(t) e^{-j\omega t} dt \right| = \left| \sum_{n=1}^{N} \sum_{i=1}^{M_n} \int_{-\infty}^{\infty} e^{-\frac{(t-t_i)^2}{2h^2}} e^{-j\omega t} dt \right|
$$

$$
\leq \sum_{n=1}^{N} \sum_{i=1}^{M_n} \left| \int_{-\infty}^{\infty} e^{-\frac{(t-t_i)^2}{2h^2}} e^{-j\omega t} dt \right| = \sum_{n=1}^{N} \sum_{i=1}^{M_n} \left| e^{-j\omega t_i} e^{-\frac{\omega^2 h^2}{2}} \sqrt{2\pi h^2} \right| \quad (2)
$$

$$
\leq \sum_{n=1}^{N} \sum_{i=1}^{M_n} \left| e^{-j\omega t_i} \right| \left| e^{-\frac{\omega^2 h^2}{2}} \sqrt{2\pi h^2} \right| = \left( \sum_{n=1}^{N} M_n \sqrt{2\pi h^2} \right) e^{-\frac{\omega^2 h^2}{2}}.
$$

Then, we can compute the upper bound of the absolute sum of the spectral higher than a certain threshold $\omega_0$ as

$$
\int_{\omega_0}^{\infty} |\hat{\lambda}(\omega)| d\omega \leq \left( \sum_{n=1}^{N} M_n \sqrt{2\pi h^2} \right) \int_{\omega_0}^{\infty} e^{-\frac{\omega^2 h^2}{2}} d\omega = \pi \left( \sum_{n=1}^{N} M_n \right) \left( 1 - \frac{1}{\sqrt{2}} \text{erf}(\omega_0 h) \right),
$$

where $\text{erf}(x) = \frac{1}{\sqrt{\pi}} \int_{-x}^{x} e^{-t^2} dt$.

Therefore, give a bound of residual $\epsilon$, we can find an $\omega_0$ guaranteeing $\int_{\omega_0}^{\infty} |\hat{\lambda}(\omega)| d\omega \leq \epsilon$, or $\text{erf}(\omega_0 h) \geq \sqrt{2} - \frac{\sqrt{2}\epsilon}{\pi \sum_{n=1}^{N} M_n}$. The proposed basis functions $\{g_d(t)\}_{d=1}^{D}$ are selected — each $g_d(t)$ is a Gaussian function with cut-off frequency $\omega_0$ and center $\frac{(d-1)T}{D}$, where $D = \lceil \frac{T\omega_0}{\pi} \rceil$. In summary, we propose Algorithm 1 to select basis functions.

---

**Algorithm 1** Selecting basis functions

---

1: **Input:** $S = \{s_n\}_{n=1}^{N}$, residual's upper bound $\epsilon$.
2: **Output:** Basis functions $\{g_d(t)\}_{d=1}^{D}$.
3: Compute $\left( \sum_{n=1}^{N} M_n \sqrt{2\pi h^2} \right) e^{-\frac{\omega^2 h^2}{2}}$ to bound $|\hat{\lambda}(\omega)|$.
4: Find the smallest $\omega_0$ satisfying $\int_{\omega_0}^{\infty} |\hat{\lambda}(\omega)| d\omega \leq \epsilon$.
5: The Gaussian basis functions $\{g_d(t)\}_{d=1}^{D}$ are with cut-off frequency $\omega_0$ and centers $\{\frac{(d-1)T}{D}\}_{d=1}^{D}$, where $D = \lceil \frac{T\omega_0}{\pi} \rceil$.

---

## 3 The Details of Learning Algorithm

### 3.1 Nested EM Framework

We consider a variational distribution having the following factorization:

$$
q(\boldsymbol{Z}, \boldsymbol{\pi}, \boldsymbol{\mu}, \boldsymbol{A}) = q(\boldsymbol{Z}) q(\boldsymbol{\pi}, \boldsymbol{\mu}, \boldsymbol{A}) = q(\boldsymbol{Z}) q(\boldsymbol{\pi}) \prod_k q(\boldsymbol{\mu}^k) q(\boldsymbol{A}^k). \quad (3)
$$

An nested EM algorithm can be used to optimize (3).

**Update Responsibility (E-step).** In each *outer iteration*, the logarithm of the optimized factor $q^*(\boldsymbol{Z})$ is approximated as

$$
\log q^*(\boldsymbol{Z})
$$
$$
= \mathbb{E}_{\boldsymbol{\pi}, \boldsymbol{\mu}, \boldsymbol{A}}[\log p(\boldsymbol{S}, \boldsymbol{Z}, \boldsymbol{\pi}, \boldsymbol{\mu}, \boldsymbol{A})] + \mathsf{C}
$$
$$
= \mathbb{E}_{\boldsymbol{\pi}}[\log p(\boldsymbol{Z}|\boldsymbol{\pi})] + \mathbb{E}_{\boldsymbol{\mu}, \boldsymbol{A}}[\log p(\boldsymbol{S}|\boldsymbol{Z}, \boldsymbol{\mu}, \boldsymbol{A})] + \mathsf{C}
$$
$$
= \sum_{n,k} z_{nk} \left( \mathbb{E}[\log \pi^k] + \mathbb{E}[\log \text{HP}(s_n|\boldsymbol{\mu}^k, \boldsymbol{A}^k)] \right) + \mathsf{C}
$$
$$
= \sum_{n,k} z_{nk} \left( \mathbb{E}[\log \pi^k] + \mathbb{E}[\sum_i \log \lambda_{c_i}^k(t_i) - \sum_c \int_0^{T_n} \lambda_c^k(s) ds] \right) + \mathsf{C} \quad (4)
$$
$$
\approx \sum_{n,k} z_{nk} \left( \mathbb{E}[\log \pi^k] + \sum_i \left( \log \mathbb{E}[\lambda_{c_i}^k(t_i)] - \frac{\text{Var}[\lambda_{c_i}^k(t_i)]}{2\mathbb{E}^2[\lambda_{c_i}^k(t_i)]} \right) - \sum_c \mathbb{E}[\int_0^{T_n} \lambda_c^k(s) ds] \right) + \mathsf{C}
$$
$$
= \sum_{n,k} z_{nk} \log \rho_{nk} + \mathsf{C}.
$$

where C is a constant, and each term $\mathbb{E}[\log \lambda_c^k(t)]$ is approximated via its second-order Taylor expansion $\log \mathbb{E}[\lambda_c^k(t)] - \frac{\mathrm{Var}[\lambda_c^k(t)]}{2\mathbb{E}^2[\lambda_c^k(t)]}$ [8]. Then, we have

$$\log \rho_{nk}$$

$$=\mathbb{E}[\log \pi^k] + \sum_i \Big(\log(\mathbb{E}[\lambda_{c_i}^k(t_i)]) - \frac{\mathrm{Var}[\lambda_{c_i}^k(t_i)]}{2\mathbb{E}^2[\lambda_{c_i}^k(t_i)]}\Big) - \sum_c \mathbb{E}[\int_0^{T_n} \lambda_c^k(s)ds]$$

$$=\mathbb{E}[\log \pi^k] + \sum_i \Big(\log(\mathbb{E}[\mu_{c_i}^k] + \sum_{j<i,d} \mathbb{E}[a_{c_ic_jd}^k]g_d(\tau_{ij})) - \frac{\mathrm{Var}[\mu_{c_i}^k] + \sum_{j<i,d}\mathrm{Var}[a_{c_ic_jd}^k]g_d^2(\tau_{ij})}{2(\mathbb{E}[\mu_{c_i}^k] + \sum_{j<i,d}\mathbb{E}[a_{c_ic_jd}^k]g_d(\tau_{ij}))^2}\Big)$$

$$\quad - \sum_c (T_n \mathbb{E}[\mu_c^k] + \sum_{i,d} \mathbb{E}[a_{cc_id}^k]G_d(T_n - t_i))$$

$$=\mathbb{E}[\log \pi^k] + \sum_i \Big(\log(\sqrt{\tfrac{\pi}{2}}\beta_{c_i}^k + \sum_{j<i,d} \sigma_{c_ic_jd}^k g_d(\tau_{ij})) - \frac{\frac{4-\pi}{2}(\beta_{c_i}^k)^2 + \sum_{j<i,d}(\sigma_{c_ic_jd}^k g_d(\tau_{ij}))^2}{2(\sqrt{\tfrac{\pi}{2}}\beta_{c_i}^k + \sum_{j<i,d}\sigma_{c_ic_jd}^k g_d(\tau_{ij}))^2}\Big)$$

$$\quad - \sum_c (T_n \sqrt{\tfrac{\pi}{2}}\beta_c^k + \sum_{i,d} \sigma_{cc_id}^k G_d(T_n - t_i)),$$

where $G_d(t) = \int_0^t g_d(s)ds$ and $\tau_{ij} = t_i - t_j$. The second equation above is based on the prior that all of the parameters are independent to each other. The term $\mathbb{E}[\log \pi^k] = \psi(\alpha_k) - \psi(\sum_k \alpha_k)$, where $\psi(\cdot)$ is the digamma function.[3] Then, the responsibility $r_{nk}$ is calculated as

$$r_{nk} = \mathbb{E}[z_{nk}] = \frac{\rho_{nk}}{\sum_j \rho_{nj}}, \text{ and } N_k = \sum_n r_{nk}. \tag{5}$$

It should be noted that here we increase $q^*(\mathbf{Z})$ via maximizing its upper bound in each iteration because the difference between $q^*(\mathbf{Z})$ and its upper bound is bounded tightly. In particular, $q^*(\mathbf{Z})$ in (4) involves $\mathbb{E}[\log \lambda_{c_i}^k(t_i)]$, which is approximated via Jensen's inequality as $\log \mathbb{E}[\lambda_{c_i}^k(t_i)]$. It actually is the first order Talyor expansion of $\mathbb{E}[\log \lambda_{c_i}^k(t_i)]$. The second order term is bounded well and the higher order terms can be ignored. We prove the rationality of our relaxation in the appendix.

**Update Parameters (M-step).** The optimal factor $q^*(\boldsymbol{\pi}, \boldsymbol{\mu}, \mathbf{A})$ is

$$\log q^*(\boldsymbol{\pi}, \boldsymbol{\mu}, \mathbf{A})$$
$$=\sum_k \log(p(\boldsymbol{\mu}^k)p(\mathbf{A}^k)) + \mathbb{E}_{\mathbf{Z}}[\log p(\mathbf{Z}|\boldsymbol{\pi})] + \log p(\boldsymbol{\pi}) + \sum_{n,k} r_{nk} \log \mathrm{HP}(\boldsymbol{s}_n|\boldsymbol{\mu}^k, \mathbf{A}^k) + \mathsf{C}. \tag{6}$$

We can estimate the parameters of Hawkes processes via:

$$\max_{\boldsymbol{\mu}, \mathbf{A}} \ \log(p(\boldsymbol{\mu})p(\mathbf{A})) + \sum_{n,k} r_{nk} \log \mathrm{HP}(\boldsymbol{s}_n|\boldsymbol{\mu}^k, \mathbf{A}^k).$$

Here, we need to use an iterative method to solve the above optimization problem. Specifically, we initialize $\boldsymbol{\mu}$ and $\mathbf{A}$ via the expectations of their distributions (used in E-step), i.e., $\boldsymbol{\mu} = \sqrt{\tfrac{\pi}{2}}\mathbf{B}$ and $\mathbf{A} = \boldsymbol{\Sigma}$. Applying the Jensen's inequality, we obtain the surrogate function of the objective function:

$$\log(p(\boldsymbol{\mu})p(\mathbf{A})) + \sum_{n,k} r_{nk} \log \mathrm{HP}(\boldsymbol{s}_n|\boldsymbol{\mu}^k, \mathbf{A}^k)$$

$$=\sum_{c,k} \Big[\log \mu_c^k - \frac{1}{2}(\frac{\mu_c^k}{\beta_c^k})^2\Big] - \sum_{c,c',d,k} \frac{a_{cc'd}^k}{\sigma_{cc'd}^k} + \sum_{n,k} r_{nk}\Big[\sum_i \log \lambda_{c_i}^k(t_i) - \sum_c \int_0^{T_n} \lambda_c^k(s)ds\Big]$$

$$\geq \sum_{c,k} \Big[\log \mu_c^k - \frac{1}{2}(\frac{\mu_c^k}{\beta_c^k})^2\Big] - \sum_{c,c',d,k} \frac{a_{cc'd}^k}{\sigma_{cc'd}^k} + \sum_{n,k} r_{nk}\Big[\sum_i \Big(p_{ii}^k \log \frac{\mu_{c_i}^k}{p_{ii}} + \sum_{j<i,d} p_{ijd}^k \log \frac{a_{c_ic_jd}^k g_d(\tau_{ij})}{p_{ijd}}\Big)$$

$$\quad - \sum_c T_n \mu_c^k - \sum_{c,i,d} a_{cc_id}^k G_d(T_n - t_i)\Big] = Q,$$

where $p_{ii}^k = \frac{\mu_{c_i}^k}{\lambda_{c_i}^k(t_i)}$, and $p_{ijd}^k = \frac{a_{c_i c_j d}^k g_d(\tau_{ij})}{\lambda_{c_i}^k(t_i)}$. Setting $\frac{\partial Q}{\partial \mu_c^k} = 0$ and $\frac{\partial Q}{\partial a_{cc'd}^k} = 0$, we have

$$\hat{\mu}_c^k = \frac{-b + \sqrt{b^2 - 4ac}}{2a}, \quad \hat{a}_{cc'd}^k = \frac{\sum_n r_{nk} \sum_{i:c_i=c} \sum_{j:c_j=c'} p_{ijd}^k}{1/\sigma_{cc'd}^k + \sum_n r_{nk} \sum_{i:c_i=c'} G_d(T_n - t_i)}. \tag{7}$$

where $a = \frac{1}{(\beta_c^k)^2}$, $b = \sum_n r_{nk} T_n$, $c = -1 - \sum_n r_{nk} \sum_{i:c_i=c} p_{ii}^k$. After repeating several such *inner iterations*, we can get optimal $\hat{\mu}$, $\widehat{A}$, and update distributions as

$$\Sigma^k = \widehat{A}^k, \ B^k = \sqrt{2/\pi} \hat{\mu}^k. \tag{8}$$

The distribution of clusters can be estimated via $\pi^k = \frac{N_k}{N}$.

## 3.2 Update The Number of Clusters $K$ via MCMC

In the case of infinite mixture model, we can apply the Markov chain Monte Carlo (MCMC) [2,10,11] to update $K$ via merging or splitting clusters.

**Chose move type.** We make a random choice to propose a combine or a split move. Let $q_m$ and $q_s = 1 - q_m$ denote the probability of proposing a merge and a split move, respectively, for a current $K$. Following the work in [10], we use $q_m = 0.5$ for $K \geq 2$, and $q_m = 0$ for $K = 1$.

**Merge move.** We randomly select a pair $(k_1, k_2)$ of components to merge and form a new component $k$. The probability of choosing $(k_1, k_2)$ is $q_c(k_1, k_2) = \frac{1}{K(K-1)}$. For our model, we can apply the following deterministic transformation to get new merged parameters:

$$\pi^k = \pi^{k_1} + \pi^{k_2}, \quad A^k = \frac{\pi^{k_1}}{\pi^k} A^{k_1} + \frac{\pi^{k_2}}{\pi^k} A^{k_2}, \quad \mu^k = \frac{\pi^{k_1}}{\pi^k} \mu^{k_1} + \frac{\pi^{k_2}}{\pi^k} \mu^{k_2}. \tag{9}$$

Then $\Sigma$ and $B$ are updated accordingly.

**Split move.** We randomly select a component $k$ to split into two new components $k_1$ and $k_2$. The probability of choosing component $k$ is $q_s(k) = \frac{1}{K}$. Different from the sampling method in previous work [2,10,11], the splitting of parameters is an ill-posed problem with positive constraints. Here, we apply a simple heuristic transformation to get new splitting parameters:

$$\pi^{k_1} = a\pi^k, \ \pi^{k_2} = (1-a)\pi^k, \ a \sim Be(1,1),$$
$$A^{k_1} = \frac{1}{2a} A^k, \ A^{k_2} = \frac{1}{2(1-a)} A^k, \quad \mu^{k_1} = \frac{1}{2a} \mu^k, \ \mu^{k_2} = \frac{1}{2(1-a)} \mu^k. \tag{10}$$

Then $\Sigma$ and $B$ are updated accordingly.

**Acceptance.** Given original parameters $\Theta$ and the new $\Theta'$, we accept a merge/split move with the probability $\min\{1, \text{likelihood ratio} \times \frac{p(\Theta')}{p(\Theta)}\}$.

## 3.3 Computational Complexity and Acceleration

Given $N$ training sequences of $C$-dimensional Hawkes processes, each of which contains $I$ events, we represent impact functions by $D$ basis functions and set the maximum number of clusters to be $K$. In the worst case, the computational complexity per iteration of our learning algorithm is $\mathcal{O}(KDNI^3C^2)$. Fortunately, the exponential prior of tensor $A$ corresponds to a sparse regularizer. In the learning phase, we can ignore the computations involving the elements close to zero to reduce the computational complexity. If the number of nonzero elements in each $A^k$ is comparable to $C$, then the computational complexity of our algorithm will be $\mathcal{O}(KDNI^2C)$. Additionally, the parallel computing techniques can also be applied to further reduce the runtime of our algorithm. Note that the learning algorithm of MMHP discretizes each impact function into $L$ points and estimates them via finite element analysis. The low-rank regularizer is imposed on its parameters. Therefore, its computational complexity per iteration is $\mathcal{O}(NI(I^2C^2 + L(C+I)) + C^3)$. Similarly, when the parameters of each Hawkes process is sparse, its computational complexity will reduce to $\mathcal{O}(NI(IC + L(C+I)) + C^2)$. The first part $\mathcal{O}(NI(IC+L(C+I)))$ corresponds to the ODE-based parameter updating while the second part $\mathcal{O}(C^2)$ corresponds to the soft-thresholding of parameters. According to the setting in [3,12], generally $L \gg I$. Therefore, the computational complexity of our algorithm is superior to that of MMHP, especially in high dimensional cases (i.e., large $C$).

(a) Random Sparse Infectivity Matrices

(b) Blockwise Sparse Infectivity Matrices

Figure 1: Comparison for various inner iteration allocation strategies on different synthetic data sets.

# 4  Experiments

## 4.1  More Experiments of Convergence

We test various inner iteration allocation strategies on two synthetic data sets. Both of these two data sets have sparse $\boldsymbol{A}$. In the first data set, the nonzero elements in $\boldsymbol{A}$ are distributed randomly, while in the second data set, each slide of $\boldsymbol{A}^k$, $k = 1, ..., K$, contain several all-zero columns and rows (i.e. blockwise sparse tensor). The convergence curves obtained via various strategies are shown in Fig. 1, which demonstrate that the open-loop control and the Bayesian optimization strategies outperform heuristic strategies consistently. The results are robust to the changes of parameters' structure and the number of clusters. According to the derivations of algorithm and the convergence analysis above, we give the scheme of our DMHP learning algorithm in Algorithm 2.

## 4.2  Synthetic Data

Fig. 2 shows the histograms of the number of clusters obtained via various methods on our two synthetic data sets ($K = 5$). We can find that the distributions obtained by our method are more concentrated to the real number of clusters.

## 4.3  Real-world Data

The clustering results of IPTV data are shown in Fig. 3. Compared with the results obtained via MMHP+DPGMM, the histogram of the number of clusters obtained via our DMHP method is more concentrated and the infectivity matrices of clusters are more structural and explainable.

## Footnotes

[2]The derivation is simple. Interested reader can try the case with $n_1 = 0, n_2 = 1, n_3 = 3$

[3]Denote the gamma function as $\Gamma(t) = \int_0^\infty x^{t-1}e^{-x}dx$, the digamma function is defined as $\psi(t) = \frac{d}{dt}\ln\Gamma(t)$.

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

**Algorithm 2** Learning DMHP

1: **Input:** $S = \{s_n\}_{n=1}^{N}$, the maximum number of clusters $K$, the maximum number of iteration $I$.
2: **Output:** Optimal parameters of model, $\hat{\alpha}$, $\hat{\Sigma}$, and $\hat{B}$.
3: Initialize $\alpha$, $\Sigma$, $B$ and $[r_{nk}]$ randomly, $i = 0$.
4: **repeat**
5:     **Just *M-step*:**
6:     Given $[r_{nk}]$, update $\{\hat{\mu}^{(1)}, \hat{A}^{(1)}\}$ via (7), calculate negative log-likelihood $L^{(1)}$.
7:     **A loop of *E-step* and *M-step*:**
8:     Given $\{\alpha, \Sigma, B\}$, update responsibility via (5), denoted as $[r_{nk}^2]$ .
9:     Given $[r_{nk}^2]$, update $\{\hat{\mu}^{(2)}, \hat{A}^{(2)}\}$ via (7), calculate negative log-likelihood $L^{(2)}$.
10:     **If** $L^{(1)} < L^{(2)}$
11:         Given $\{\hat{\mu}^{(1)}, \hat{A}^{(1)}\}$, update $\Sigma$, $B$ via (8).
12:     **Else**
13:         Update $[r_{nk}]$ via $[r_{nk}^{(2)}]$.
14:         Given $[r_{nk}], \hat{\mu}^{(2)}, \hat{A}^{(2)}$, update $\alpha$, $\Sigma$, $B$ via (8).
15:     **End**
16:     Merge or split clusters and update $\Sigma$, $B$ via MCMC.
17:     $i = i + 1$.
18: **until** $i = I$
19: $\hat{\alpha} = \alpha$, $\hat{\Sigma} = \Sigma$, and $\hat{B} = B$.

(a) Sine-like impact function     (b) Piecewise constant impact function

Figure 2: The histograms of the number of clusters obtained via various methods on the two synthetic data sets.

[4] E. Meijer and J. Y. Ypma. A simple identification proof for a mixture of two univariate normal distributions. *Journal of Classification*, 25(1):113–123, 2008.

[5] T. J. Rothenberg. Identification in parametric models. *Econometrica: Journal of the Econometric Society*, pages 577–591, 1971.

[6] B. W. Silverman. *Density estimation for statistics and data analysis*, volume 26. CRC press, 1986.

[7] A. Simma and M. I. Jordan. Modeling events with cascades of Poisson processes. In *UAI*, 2010.

[8] Y. W. Teh, D. Newman, and M. Welling. A collapsed variational Bayesian inference algorithm for latent Dirichlet allocation. In *NIPS*, 2006.

[9] H. Xu, M. Farajtabar, and H. Zha. Learning Granger causality for Hawkes processes. In *ICML*, 2016.

[10] Y. Xu, P. Müller, and D. Telesca. Bayesian inference for latent biologic structure with determinantal point processes (DPP). *Biometrics*, 2016.

[11] Z. Zhang, K. L. Chan, Y. Wu, and C. Chen. Learning a multivariate Gaussian mixture model with the reversible jump MCMC algorithm. *Statistics and Computing*, 14(4):343–355, 2004.

[12] K. Zhou, H. Zha, and L. Song. Learning triggering kernels for multi-dimensional Hawkes processes. In *ICML*, 2013.

(a) Histogram of $K$

(b) DMHP

(c) MMHP+DPGMM

Figure 3: Comparisons on the IPTV user data.

(a) MMHP+DPGMM

(b) DMHP

Figure 4: Comparisons for various methods on F1 score of minor cluster.