[Reviews · NeurIPS 2017]

Reviewer 1



This paper considers the problem of clustering event sequences. To this end, the authors introduce a Dirichlet mixture of Hawkes processes and derive a variational-EM algorithm for its inference. The work is sound and may be useful in some contexts but, in my opinion, not enough innovative to reach the acceptance level. In addition, the use of MCMC within a variational framework for updating the number of clusters is surprising and can slow down the inference algorithm. Furthermore, the application on real data is somehow disappointing since the competitors seem unfavored.

Reviewer 2



In this work, the authors introduce a novel method to cluster sequences of events. This method relies on a Dirichlet mixture model of Hawkes processes to describe the data and assign events to clusters. This differs from previous work using Hawkes Processes for sequential data by focusing directly on clustering the sequences themselves rather than sub-events; in particular, compared to prior work such as by Luo et al., 2015, the clustering is done as part of the modeling itself rather than applied after learning the parameters of the model, which leads to a sizable improvement in experimental results. Quality: This paper is of good quality. Clarity: This paper is overall clear, but contains some grammar and style mistakes - some have been noted below in the detailed comments, but the authors ideally should have their manuscript proof-read for English grammatical and style issues. Figure 2 is too small to be easily readable. Originality: This paper introduces a new method to model sequences of events. Significance: This paper appears to be significant. Detailed comments: - In section 2, it would be valuable to provide a formal definition of Hawkes Process; in particular, it would improve readability to put the intensity equation (line 67) in its own equation environment, and then refer to it by number in Section 3 when relevant. - When comparing line 67 to Eq. 1, it seems that the sum over the event types (c_i) is missing; could you please clarify this? - Line 156: you state that the runtime of the algorithm is linearly proportional to the number of inner iterations - could you write out the complexity explicitly? - Along the same lines, could you please state the complexity of MMHP in section 4.4? - The second paragraph of section 4.2 needs to be made clearer. Clarity comments: - You do not need to define R+ line 101 - line 23: "the previous two" - line 39: "prior of the clusters" - line 51: "the problems of overfitting" - line 61: "A Hawkes" - line 73: "was first proposed" - line 77: "was used" - line 100: "basis" - line 109: "a lot of work" should be rephrased - line 125: "goes to infinity" - line 232: "state of the art" - line 255: spacing

Reviewer 3



This paper proposes an interesting idea for event sequence clustering. It made some assumptions on the clustering behavior across sequences and applied a mixture of Hawkes processes. Properties of the model including local identifiability, sample complexity and computation complexity are analyzed. Experimental results are provided with some baselines, demonstrating the modeling power of the proposed method. 1. At the end of Sec 3.1, the comment in the last sentence is a bit ambiguous. If I understand correctly, this model is a mixture model rather than an admixture model like LDA. Once a cluster label is selected for a sequence, the events are generated via a single Hawkes process instead of a mixture. Am I misunderstanding something or can you clarify the comment? 2. In the synthetic data experiments, the results indeed show the algorithm can handle certain degree of model misspecification. But that misspecification is rather minor, in terms of how good a function can be fit in a different function space. However, the data is generated with the same clustering assumptions proposed by the authors. If the clustering assumption is consistent with some baseline method, e.g., VAR+DPGMM, how would you expect the DMHP performance will be? 3. On the inferred cluster number distribution in the synthetic data, two examples are provided in the supplements but the ground truth is not described there. Is it always concentrated around the ground truth as the cluster number varies from 2 to 5?